# Conserved microRNAs and Flipons Shape Gene Expression during Development by Altering Promoter Conformations

**DOI:** 10.3390/ijms24054884

**Published:** 2023-03-03

**Authors:** Alan Herbert, Fedor Pavlov, Dmitrii Konovalov, Maria Poptsova

**Affiliations:** 1InsideOutBio, 42 8th Street, Charlestown, MA 02129, USA; 2Laboratory of Bioinformatics, Faculty of Computer Science, National Research University Higher School of Economics, 11 Pokrovsky Bulvar, 101000 Moscow, Russia

**Keywords:** flipons, Z-DNA, G4 Quadruplex, H-DNA, SIDD, microRNA, embryonic development, promoter, retroelement, transcription, histone marks, gene expression, epigenetics, glutamate receptors, synapses

## Abstract

The classical view of gene regulation draws from prokaryotic models, where responses to environmental changes involve operons regulated by sequence-specific protein interactions with DNA, although it is now known that operons are also modulated by small RNAs. In eukaryotes, pathways based on microRNAs (miR) regulate the readout of genomic information from transcripts, while alternative nucleic acid structures encoded by flipons influence the readout of genetic programs from DNA. Here, we provide evidence that miR- and flipon-based mechanisms are deeply connected. We analyze the connection between flipon conformation and the 211 highly conserved human miR that are shared with other placental and other bilateral species. The direct interaction between conserved miR (c-miR) and flipons is supported by sequence alignments and the engagement of argonaute proteins by experimentally validated flipons as well as their enrichment in promoters of coding transcripts important in multicellular development, cell surface glycosylation and glutamatergic synapse specification with significant enrichments at false discovery rates as low as 10^−116^. We also identify a second subset of c-miR that targets flipons essential for retrotransposon replication, exploiting that vulnerability to limit their spread. We propose that miR can act in a combinatorial manner to regulate the readout of genetic information by specifying when and where flipons form non-B DNA (NoB) conformations, providing the interactions of the conserved hsa-miR-324-3p with *RELA* and the conserved hsa-miR-744 with *ARHGAP5* genes as examples.

## 1. Introduction

The approach we take follows that of Britten and Davidson, who first proposed the programming of genomes by *trans*-RNAs that act at locations distant from where they are produced [1]. While there are many classes of *trans*-RNAs [2], we will focus on miR and examine the DNA sites to which they bind, and their potential role in embryonic development. We will investigate the interactions between miR and flipons, sequences that form NoB under physiological conditions [3]. We show that one type of flipon is targeted by c-miR to suppress the expression of endogenous retroelements (ERE). We also show that a different set of c-miR target flipons in developmentally important promoters within the candidate *cis*-regulatory elements (cCRE) previously identified by the Encyclopedia of DNA Elements (ENCODE) consortium. The promoters are enriched in genes that encode homeobox and other proteins essential for the specification of embryonic tissues. Interestingly, genes associated with cell-membrane-mediated communications, synapse formation and glutamatergic signaling are also targets for c-miR [4].

## 2. The Links between miR and Gene Regulation

The ability of miR to regulate phenotype was first shown by effects arising from the suppression of RNA translation during development. By changing the time window in which a protein is produced, the miR altered the size and cellular composition of tissues [5]. The miR arise through a number of canonical and non-canonical processing pathways that target AGO1 and AGO2 argonaute effector proteins to transcripts with base-complementarity to the miR seed sequence (mRS) [6]. In line with a role for *trans*-RNAs in these early outcomes, *Ago2* knockout in mouse embryos is embryonically lethal, with a failure to develop extraembryonic tissue. Surprisingly, survival does not depend on the AGO2 protein’s enzymatic slicer activity to suppress transcript translation, as mice homozygous for a catalytically dead *Ago2^ADH^* allele develop normally [7], suggesting that AGO2 has other roles. In addition to the cytoplasm, miR also localize to the nucleus [2,8]. They modulate gene expression [2,9,10] and potentially some alternative splicing events [11,12,13]. They also have effects on gametogenesis [14]. Small RNAs also repress expression of ERE in germline tissue and embryos. In addition to miR, P-element-induced wimpy testis protein (PIWI)-associated RNAs (piRNA) and transposon-derived small-interfering RNAs (siRNA) play a role in ERE suppression [15,16]. Small synthetic RNAs (saRNA) can also activate gene expression in an AGO2-dependent fashion. The first saRNA was unexpectedly discovered by tiling of the progesterone receptor (encoded by *PGR*) promoter with siRNA mimetics, excluding regions with either high GC content or low sequence complexity [17,18]. Later, the p21 (encoded by *CDKN1A*) promoter [19] and the CCATT/enhancer binding protein alpha (encoded by *CEPBA*) were also effectively activated by saRNA generated empirically [20]. In each of these three cases, the target site in the promoter of each gene is within a region of antisense transcription (from *PGR-AS1* for *PGR*, *DINOL* for *CDKN1A* and the polycomb-associated non-coding RNA ENST00000587312.1 for *CEPBA*). However, targeted destruction of PGR-*AS*1 with siRNAs does not diminish saRNA-induced gene expression [17], while siRNAs against the anti-sense transcript increase anti-sense *CEBPA* RNA transcription [20]. The *PGR* saRNA binds within an *LIMB8* LINE (long-interspersed nuclear element) while the CEPB saRNA site lies within a CpG island, raising the question of whether such classes of repeat sequences are also targeted by cellular small RNAs to regulate gene expression.

## 3. Flipons

Flipons are another example of unconventional gene regulation. They are often composed of simple repeats that have little informational value because of their high frequency in the genome. Instead, these genetic elements signal by shape rather than by sequence [21]. They adopt an alternative NoB under physiological conditions in regions subject to topological stress (Figure 1) [22]. The flip to left-handed Z-DNA is favored by an alternating pyrimidine (Y) and purine (R) (d(YR)_n_) consensus sequence (CS), while quadruplexes (G4) composed of G quartets have a CS of d(G_3-5_X_1-7_)_4_ (X is any nucleotide), and triplexes have a CS of d(R)_n_ (Figure 1A) [23]. Interestingly, NoB can form with non-canonical basepairs [24,25], while a four-stranded Z-quadruplex has recently been described [26]. The flip to NoB can be powered by polymerases and processive helicases [22,27,28]. The transformation occurs faster than topoisomerases can relax the DNA torque produced [29]. A role for flipons in gene expression has been proposed almost since the discovery of these non-B-DNA structures and there is significant suggestive evidence that they do so [30,31]. Indeed, the first genome sequences revealed enrichment of Z and G flipons in promoters [32]. It is likely that the NoB they form flag these regions for easy discovery by the cellular machinery controlling gene readout. The NoBs further ensure that transcription start sites (TSS) are accessible since neither nucleosomes nor B-DNA-specific proteins bind with high affinity to these alternative conformations [21,33,34,35].

The roles specific for each different flipon type likely reflect differences in the kinetics of their formation and resolution, with G4 more stable than Z-DNA [31]. Folding of a triplex is more complex and requires opening up of the major groove of the purine-rich sequences to permit docking of a third strand. Triplexes can form locally by the fold-back of a single-stranded region of polypurine or polypyrimidine DNA onto an adjacent sequence-matched duplex to form H-DNA [36]. S flipons (CS d(ATX)_n_) are sites of stress-induced duplex destabilization (SIDD) [37], where the accumulation of energy drives unpairing of the helical strands to form loops. Importantly, SIDDs differ from the RNA-induced DNA loops (called R loops) produced within gene bodies in GC-rich regions during transcription or as a result of replication stress [38,39]. In contrast to such R loops, S flipons do not overlap the RNA-polymerase peaks detected by ChIP-seq experiments [36]. S flipons are likely to arise in the genome during retrotransposon events involving long terminal repeats (LTRs) and LINEs (collectively transposon repeat elements (TREs)) (Figure 1B). In addition to RNA or DNA, SIDDs can bind KH-domain proteins like those regulating c-MYC gene expression that have high affinity for single-stranded DNA [40] or those involved in alternative RNA splicing [41].

## 4. Flipons and miR

Here, we ask whether these two systems of unconventional gene regulation interact. Genome-wide flipon maps generated by the *in cellula* nucleotide resolution data from potassium permanganate (KMnO_4_)/S1 nuclease footprinting [36] allow us to test whether experimentally validated NoBs are targeted by miR, as shown in Figure 1C for G flipons and Figure 1D for Z flipons. We base our analysis on c-miR families, some of which trace back to the earliest bilateral ancestor. Many of these c-miR have important functions in metazoan development [6]. Here, we identify roles for flipons in these processes. We show that G and Z flipons cluster around TSS in around 33% of coding genes and that these sequences are targeted by c-miR in around 18% of coding gene proximal promoters. The findings are consistent with a model in which the combination of miR expressed in a cell regulates the flipon conformations within genes specifying cell-to-cell communications, mediated both by carbohydrates and neurotransmitters, processes that ultimately determine cell fate.

The interactions between flipons and c-miR allow for rapid and dynamic changes in RNA expression during development or in response to environmental perturbations by determining the cellular machines that dock at TSS and the chromatin modifications that result. Our findings support a scenario in which the genome comprises an evolving canvas that is framed by flipons, modeled by miR, particularized by proteins and endlessly embellished through the natural selection of synergies.

## 5. Results

We mapped experimentally validated flipons present in the mouse genome (mm10) (Figure 2A), annotating sites for the presence of cCRE and TRE (Appendix A). We found enrichment of flipons in proximal promoters (PP, less than or equal to 1 kb from TSS), most notably for G and Z flipons (Figure 2, Figure 3, Figure 4, Figure 5 and Figure 6). The PP flipons were associated with cCRE marks. Flipons, particularly SIDDs, also localized to distal intergenic regions and overlapped TRE (Figure 2 and Figure 4) that lack cCRE.

We then mapped evolutionarily conserved miR seed sequences (c-mRS) from placental animals (Appendix A) [6] to test for their interaction with experimentally defined flipon motifs (Figure 2A,B and Appendix A), noting the number of interactions of miR per flipon type along with the strand preference (Figure 2C), how frequently c-miR bound to flipon junctions rather than to motifs (see Figure 1E,F and Figure 6A, Appendix A). We further detailed the location of flipons and c-miR in PP (Figure 3, Appendix A), whether different c-miR map to PP and TRE (Figure 4), the relative proportion of mRS in coding and noncoding PP (Appendix A) and whether there is enrichment of argonaute binding sites in PP as would be expected if the proteins were guided there by c-miR (Figure 6 and Appendix A). We then undertook genome-wide annotation of the c-miR interactions with flipons in cCRE and TRE regions (Figure 3 and Appendix A) and tested whether genes in particular biological pathways were enriched (Appendix A).

### 5.1. SIDDs and miR

For flipons present in distal intergenic regions, mRS overlapped mainly with the AT-rich SIDDs (mean length = 170 bp). The most frequent overlap was for miR-203a (~20%) (Figure 4). Other miR matches were also common, including for miR-374 that is encoded within the X-inactivation center and linked to many pathologies [42] and miR-186 that is most often a tumor suppressor [43]. Many other miR also preferentially target SIDDs but at lower frequency (Figure 2A). Overall, SIDDs matching mRS lacked cCRE marks.

### 5.2. Promoter Flipons and miR

In contrast, proximal promoters (PP) within 1 kb of a TSS were enriched for cCRE marks and cCRE marks associated with CCCTC binding factor (CTCF). Z and G flipons (Figure 2A), but not S and T flipons, were enriched in PP with the total number of sequence matches to c-mRS occurring more frequently for Z flipons than for G flipons (Figure 2B and Appendix A).

### 5.3. Strand Preference for c-mRS Matches

There was no apparent strand preference for miR binding to flipons, either for TRE or for PP, with respect to the direction of transcription (Figure 2C). When analyzed by binding to DNA strands, the total number of sites with a c-miR match for a particular flipon was much larger for TRE than for promoters, an outcome that may, in part, reflect the larger overall size of S flipons (mapping is at a resolution of ~170 bp [44]).

**Figure 5 ijms-24-04884-f005:**
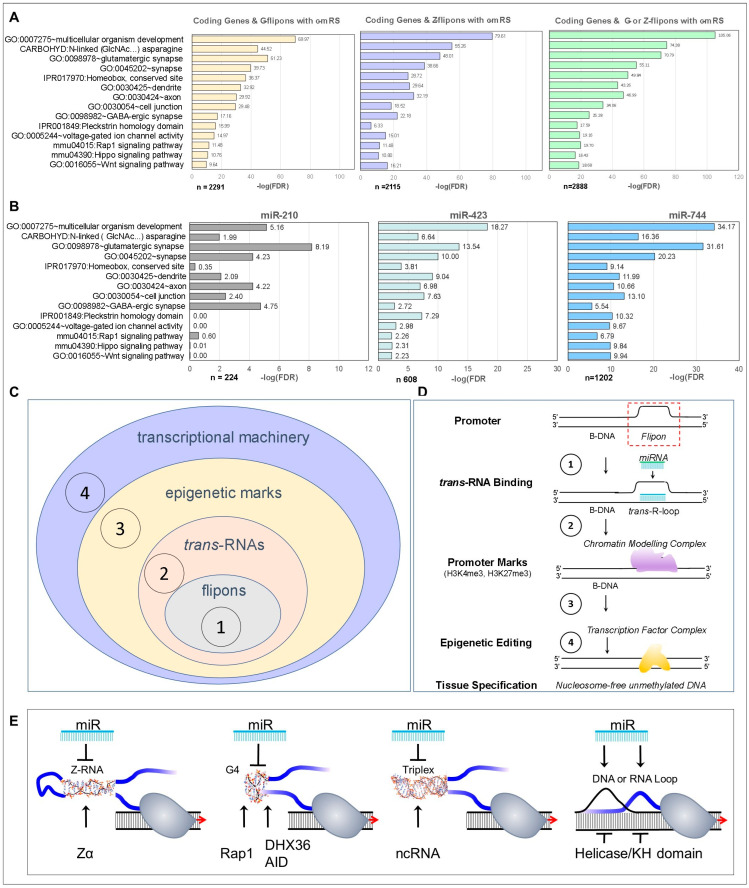
Flipons and gene expression (**A**) GO pathways enriched for G and Z flipons with c-mRS in PP, showing G and Z flipon types individually or after the two gene lists were merged. (**B**) GO analysis for individual c-miR from Figure 4. Please note that the X-axis is different for each miR. (**C**,**D**) A model where the interaction between miR and flipons sets promoter conformation. The NoBs formed in the region then can localize chromatin modeling complexes through structure-specific domains. Outcomes will also depend on which regulatory factors the different argonaute family members engage. While the small RNAs provide specificity, the NoB catalyze the assembly of complexes through the release of potential and chemical free energy that arises either directly from the flip of NoB back to B-DNA or from the activation of enzymes like helicases that cleave nucleotide triphosphates. (**E**) The various ways in which miR can interact with nascent transcripts. Protein motifs modulating flipon and RNA conformation include the Zα domain [45], yeast Rap 1 [46], DHX36 [47], AID [48], KH domains [41], helicases and topoisomerases (not shown) (FDR: False Discovery Rate; GO: gene ontology; PP: proximal promoter ≤ 1kb from a TSS; TSS: transcription start site).

### 5.4. Interaction of Individual c-miR by Flipon Type, Genomic Region and DNA Strand

The number of interactions with flipons for individual c-miR varied greatly (Figure 2, Figure 3 and Figure 4). The number of genes with c-mRS was highest for miR-147 and miR-210 that bound Z flipons and for miR-296, miR-423, miR-491 and miR-532 that are specific for G flipon. Interestingly, miR-744 had matches for both G and Z flipons (Figure 4). Overall, the counts for each miR were similar on both template and coding DNA strands. The interactions with G and Z flipons in PP frequently overlapped cCRE regions. In contrast, counts for SIDD-interacting c-miR were highest in TRE regions, with miR-203a being the most common. The mRS seed sequence matches were not often associated with cCRE.

### 5.5. Colocalization of G and Z Flipons in Promoters

To examine the role of G and Z flipons in PP, we mapped their location in 100 base pair windows either side of the TSS (Figure 3A and Appendix A). We found that G and Z flipon counts were highest around the TSS (Figure 3B), as were the sites bound by c-miR. The clustering around TSS is consistent with past selection to enrich for these flipon–miR interactions. We observed a peak of G flipons bound by c-miR in the −100 bp window (Figure 3C), while the Z flipon with c-mRS distribution was skewed towards the transcribed windows, where negative supercoiling due to transcription first arises (Figure 3D). Overall, the findings indicate that G and Z flipons perform different roles in PP. The presence of multiple flipons in a subset of PP, many also bound by c-miR, increases the complexity of possible outcomes by exploiting the unique biology of each flipon type (Figure 3E,F and Appendix A).

### 5.6. Gene Ontology (GO) of Promoters

We examined GO annotation for genes whose promoters had G or Z flipons matching mRS (Appendix A), regardless of the number of PP matches observed for each gene. The enrichment of both G and Z flipons in PP was particularly high for developmental and synapse genes, with FDR < 10^−20^ in many cases (Figure 3G,H and Appendix A). Genes associated with glutamatergic synapses were enriched for flipons having c-miR matches (FDR < 10^−63^), while scores for genes specifying GABAergic synapses were less significant (FDR < 10^−18^). We also analyzed genes with G and Z flipon mRS matches separately (Figure 5A and Appendix A). Interestingly, many G and Z flipons map to different genes within the same GO annotation. Intriguingly, both G and Z flipons are enriched in homeobox gene promoters.

### 5.7. Site Selection of c-mRS in Promoters by Flipon Type

We analyzed the sites in flipons with mRS by grouping them either into motif-binding (M), junction binding (J), or as both motif and junction binding (MJ). The J sites were defined by a one base overlap with an experimentally determined flipon site, while MJ-sites required a 2-6-base overlap with the flipon sequence (Figure 6A, inset and Appendix A). We found that c-mRS corresponded mostly to M and MJ sites, although there were the following differences between flipon classes. S flipon sites were mostly M, in part reflecting their 170 bp average size. G flipon sites were more often M while Z flipons were most commonly MJ, suggesting some selection for this outcome. Overall, c-miR showed a preference in site binding that was absent for other annotated miR (Figure 6B). While c-miR target only a portion of flipons, the entire set of G and S flipons were targeted by the combination of conserved and non-conserved miR. A small portion of H and Z flipons remained untargeted by any of the miR we tested (Figure 6B and Appendix A).

**Figure 6 ijms-24-04884-f006:**
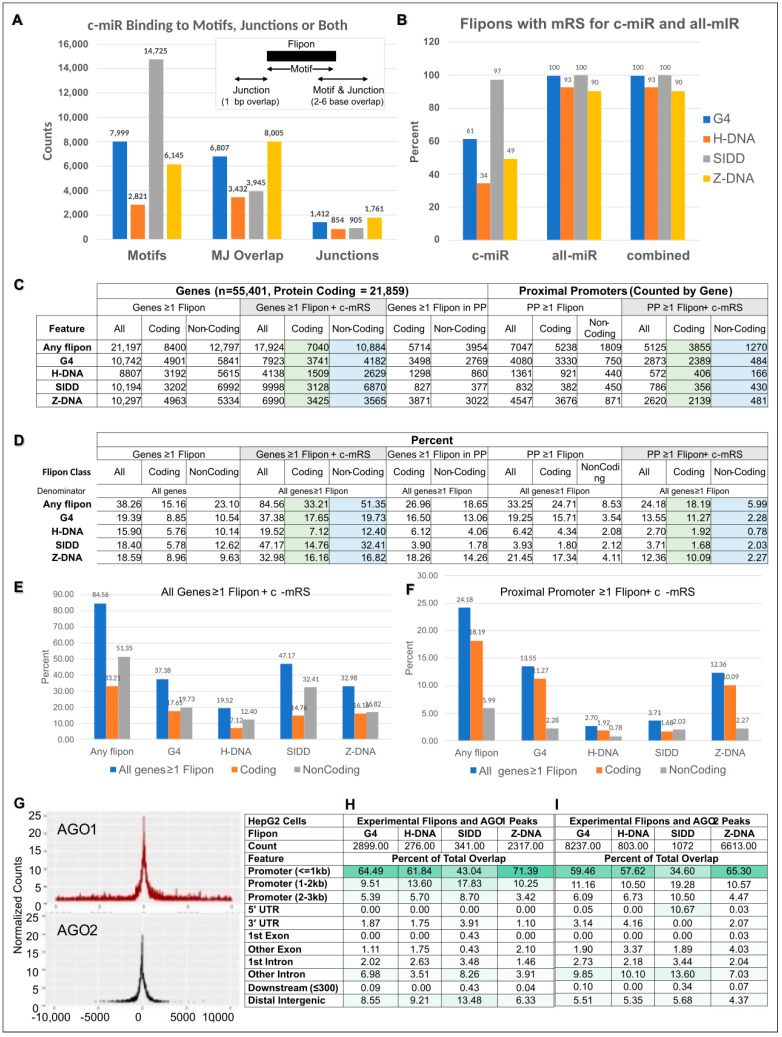
Evidence for selection of flipon interactions with c-miR at PP. (**A**) In PP, the frequency with which c-miR interact with flipon motifs (M), or with the junctions (J) between flipons and B-DNA or with both motifs and junctions (MJ) differs by flipon class. (**B**) c-miR are more selective in their interactions compared to other classes of mouse miR. (**C**) Differences also exist in the frequency with which c-miR bind PP of coding transcripts (highlighted in green) compared to non-coding transcripts (highlighted in blue). (**D**) c-miR also interact preferentially with flipons in promoters for coding compared to non-coding transcripts, a difference that was not apparent when counts were made for the entire gene body. (**E**,**F**) The differences between genes and PP are shown graphically by plotting the data from the table for cells highlighted in color. (**G**) Properties of flipons bound by the miR effector proteins AGO1 and AGO2 in human HepG2 cells. The figures show the position of ChIP-Seq peaks relative to the TSS. (**H**,**I**) In the tables, the overlaps between flipons, AGO1 and AGO2 proteins are mapped to genomic features. Binding of both proteins is enriched in PP as evidenced also by the rows highlighted in dark green (AGO: argonaute protein; TSS: transcription start site; c-miR: conserved miR; c-mRS: conserved miR seed sequence perfect 7-mer match; PP: proximal promoter ≤ 1 kb from TSS).

### 5.8. Enrichment of Flipons in Proximal Promoters of Coding Genes

We also found that flipons were enriched in proximal promoters of coding genes compared to non-coding genes (24.71% c.f. 8.53%). In contrast, the count of flipons over all gene bodies was highest for non-coding genes (Figure 6C,D) (23.10% c.f. 15.16%). This finding was also true for the location of flipons with c-mRS (tabulated in Figure 6C,D and displayed in Figure 6E,F, Appendix A).

### 5.9. Colocalization of Argonaute Protein to Promoter Flipons

To confirm the potential for interaction of miR with flipons, we examined the binding of AGO1 and AGO2 proteins to DNA using the ENCODE ChIP-seq (chromatin immunoprecipitation and sequencing) datasets that are available for the human HepG2 liver cell line, as no paired datasets were available from murine studies (Appendix A). The binding of AGO1 and AGO2 was also highest in PP (Figure 6G). Around 15–20% of G and Z flipons were within AGO1 and AGO2 peaks, with many more Z and G flipons bound than S or H flipons (Appendix A). It is possible that the S or H flipons detected in the ChIP experiments were passengers incidentally pulled down in the 150 bp promoter fragments analyzed in these studies.

### 5.10. GO analysis of Promoter-Specific miR

We performed GO annotation of those c-miR most frequently found to interact with flipons (Figure 5A,B and Appendix A). Each c-miR was associated with different gene sets. The Z-specific miR-210 mapped to genes involved glutamatergic and GABAergic synapses. The G flipon-specific miR-423 targets included genes involved in synapses, dendrites and with pleckstrin homology domains. miR-744 that can interact with G and Z flipons also targets glutamatergic synapses. The c-mRS also has matches to promoters for genes in the Wnt and Hippo signaling pathways. 

### 5.11. Modeling miR Targeting of Promoter Flipons

We noted that the potential sites at which c-miR dock to flipons are associated with cCRE marks (Figure 4). These findings are consistent with the model presented in Figure 5C,D in which chromatin modification complexes are localized by the interaction of c-miR with flipons. The resulting chromatin marks then modulate transcription factor binding. Engagement of DNA by miR would depend on the miR expression level in a cell and how many perfectly matched mRS 7-mer sites are available. We therefore counted the number of different DNA heptamers in the mouse genome. We found that all possible heptads are present in proximal promoters, but the counts range from 1 to over 150,000 indicating that in many cases the stoichiometry between c-miR and mRS matches can be quite favorable (Appendix A). However, the number of available mRS matches in a nucleus will be lower than this count, as each heptad sequence is potentially masked in a variety of ways. For example, a heptad may be buried within the DNA double helix or cloaked by chromatin. In these cases, the actual c-miR to mRS match ratio increases. Nascent transcripts within gene bodies can also compete with c-miR for matching mRS, as illustrated for the different flipon classes in Figure 5E. In such situations, it is likely that the structures formed localize protein complexes that primarily modify transcripts or respond to DNA damage. Nascent transcripts may also bind c-miR directly, increasing the local concentration of c-miR [49] and potentially creating a feedback loop that regulates the conformation of nearby flipons.

### 5.12. Architecture of Flipon and miR Interactions in Promoters

The localization of both G and Z flipons to promoters opens up the possibility of combinatorial control of gene expression by miR, as illustrated in Figure 1F. To understand this outcome, we analyzed the distribution of flipon c-mRS matches in the PP from a number of genes (Figure 7, Figure 8, Figure 9, Figure 10 and Figure 11).

### 5.13. Flipons in the Il11 Promoter

The interleukin 11 (*Il11*) gene that encodes a protein with a role in female fertility and inflammation-induced pulmonary fibrosis [51] has a large number of experimentally validated PP flipons (indicated by black rectangles) with c-mRS matches (Figure 7). The colored boxes around each flipon indicate which strand forms G4, using the same color scheme for the boxes as in Figure 1 and Figure 7. The miR labels are similarly colored i.e., if the label color is the same, then the G4 forming sequence and the c-mRS are on the same DNA strand, while contrasting colors indicate that the sequence match for each is on the opposite DNA strand. The formation of G4 is promoted only when both G4 boxes and c-miR labels have the same color (see Figure 1C and 7 for details). For *Il11* PP, the flipon and c-mRS matches favoring G4 formation are associated with histone H3 lysine 4 trimethylation marks (H3K4me3), marks that are characteristic of active promoters (Figure 7). Sites where the c-miR and flipon colors differ do not favor G4 formation and lack such marks.

### 5.14. Flipons in the Jag2 Promoter

We also analyzed the *Jag2* gene promoter (Figure 8). Jag2 encodes a protein that mediates lateral inhibition through the notch pathway, a process essential for establishing the sharp boundaries during development that specify cell fate. The PP is enriched for Z flipons with sequence matches for c-mRS. Many of these c-miR likely disrupt Z-DNA formation, not by directly interacting with a d(YR)_n_ repeat but rather by inhibiting formation of the junction between B- and Z-DNA helices (BZJ). The variable nature of BZJ forming sequences allows miR to specifically target a particular Z flipon. Instead of flipping B-DNA to Z-DNA, the miR likely promotes the opening up of the helix to expose single-stranded DNA. In the case of miR-744, the inhibition of Z-DNA will favor G4 formation. Otherwise, formation of Z-DNA will reduce the available free energy for G4 formation. The *Jag2* PP also contains two SIDDs with mRS matches to multiple c-miR. They lack cCRE marks and are not associated with TRE.

### 5.15. Flipons in the Acvr2b Promoter

The activin A receptor type 2B (Acvr2b) is a high-affinity ligand for activin, a member of the transforming growth factor superfamily that signals through the SMAD pathway and plays key roles during neural development [52,53]. The PP is enriched for Z and G flipons with c-mRS (Figure 9). The interactions of miR-296 and miR-744 mRS and G4 forming sequence are on the template strand. They promote G4 formation and are associated with cCRE promoter marks (Figure 9B,C). A Z flipon is also recognized by miR-744, with the sequence match overlapping the BZJ (Figure 8B). Here, miR-744 inhibits Z-DNA formation while promoting G4 formation. Three other Z flipons are uniquely bound by a c-miR (Figure 8B–H). All three c-miR interactions inhibit Z-DNA formation, regardless of whether the upper or lower DNA strand is bound.

### 5.16. Flipons in the Adgrb2 Promoter

The adhesion G-protein-coupled receptor B2 (*Adgrb2*), which has roles in neurogenesis [54], is enriched for G flipons in the PP and shows some of the complexities possible through combinatorial miR interactions. The promoter region is bound by three miR (Figure 10A). Only miR-491 promotes G4 formation on the coding strand while miR-296 and miR-504 inhibit this fold (Figure 10B). Other G flipons in the body of the gene, such as miR-330, favor G4 formation on the template strand and can potentially inhibit RNA polymerase elongation of the transcript. The mRS sites then allow for transcriptional regulation by miR-330 (Figure 10C) and by other miR that are less conserved across species. In each case, transcription could either be terminated to produce RNAs undergoing rapid degraded or paused to reduce the overall rate of RNA production. The promoter region is also characterized by the nucleosome-free region (NFR) that is mapped at nucleotide resolution through hydroxyl radical footprinting with a cysteine-modified histone H4 (the marks in the track labeled nucleosome indicate the location of the nucleosome dyad axis [50]). The NFRs are enriched for Z flipons that function to keep the region open and accessible to transcription factors.

### 5.17. Flipons in the Grb10 Promoter

In other situations, Z flipons have different roles to play. Their interaction with miR may prevent Z formation as described above for the *Acvr2b* promoter. For the growth factor receptor-bound protein 10 (*Grb10*) gene, there is a series of Z flipons on the coding strand, and they are self-complementary and that could potentially fold into hairpin structures (Figure 11A–D). We examined how binding of miR-187 and miR-210 affect duplex formation by focusing on the mRS matches to the DNA sequence (denoted in the figure by “(s)” for seed appended after the miR name) and on the complementary site to which the c-miR binds (denoted by “(c)” written after the miR name). Neither of the miR “(c)” interactions disrupt the alignment of the potential coding strand duplex nor prevent the flip to Z-DNA as both miR bind to the DNA template strand. With other genes, an interaction with miR can also promote the formation of an alternative Z-DNA prone structure. Another example is found in the nucleolar protein 4-like (*Nol4l*) gene where one Z flipon strand is GT rich (Figure 1D and Figure 11E,F). If we allow wobble G:T base pairs, a pairing that is fully compatible with Z-DNA formation [24], the coding strand can form a hairpin with the potential to form Z-DNA (Figure 11G), whereas the template strand cannot (Figure 11H). Binding of both miR-329/362 and miR-376c seeds to the DNA complement (indicated by “(c)” in the figure) is permissive to the coding strand hairpin fold. The duplex structure lacks a miR-210(c) site (Figure 11E,F). The *Nol4l* coding strand could instead form a left-handed quadruplex similar to the recently described fold formed by r(GT)_n_ repeats longer than 11 bases that is likely to be a more stable structure than Z-RNA [26]. We note that hairpin folds formed by *Grb2* and *Nol4l* transcripts would not be disrupted by c-miR described above and could fold as Z-RNA or potentially as a Z-quadruplex.

## 6. Discussion

We observe a high frequency of matches between c-miR and flipons and evidence that these interactions are under selection. There is enrichment of flipons in proximal promoters (PP, less than or equal to 1 kb from a TSS), most notably for G and Z flipons. The PP flipons are associated with cCRE marks and their distribution differs by flipon type. A different subset of flipons, particularly SIDDs, localize to TRE in distal intergenic regions (Figure 2A,B) that lack cCRE. As we also reported previously, both PP and TRE flipons have matches to seed sequences of c-miR families found in placental animals [6]. G and Z flipons are preferentially targeted by c-miR in PP of coding genes, while the S flipons targeted are most associated with TRE. Flipons are noticeably absent from 3′UTRs, the major site used by miR for regulating transcript stability and translation.

The function of genes with PP flipons is highly enriched for multicellular development, cell-surface N-linked glycosylation, glutamatergic and GABAergic signaling, all of which play an important role in neural development [55] and in rapid responses to environmental change (Figure 3G,H). The proteins produced affect development either through cell-surface glycosylation that generates an untemplated combinatorial sugar code to regulate cell contacts [56],via synapses and gap junctions that create electrical potentials across tissues to program embryogenesis [57], or through molecules such as phosphoinositides that establish complex chemical gradients to direct cell movements [58]. The list of enriched genes that contain c-mRS matches includes homeobox genes, transcriptional regulators and gene products expressed on outwards facing membranes (Figure 5A). Each c-miR targets a subset of flipons involved in these processes (Figure 5B), consistent with the specific phenotypes arising from the germline knockout of individual miR genes in mice [6].

Regulation of flipon conformation by c-miR is one way to control expression during embryogenesis that does not rely on the production of sequence-specific transcription protein factors. The high degree of transcription observed in the early embryo delivers the energy needed to power these flipon-dependent transactions [59]. The conditions are ideal for generating NoBs. Many of the active promoters in mouse embryonic stem cells are bidirectional and divergent with the TSS for sense and antisense transcripts often separated by less than 1 kb [60,61]. The arrangement maximizes the accumulation of negative supercoiling in PP. Channeling of the free energy by the small RNAs transmitted by parental gametes impacts promoter shape by changing flipon conformation. The process depends on the sequence-specificity of each of these RNA and the manifest present in each cell sets promoter shape. The outcome is guided by protein complexes that localize to NoB through structure-specific binding domains. The assembly of cellular machines is powered by the potential energy stored in NoB along with the chemical energy released when helicases unwind these alternative conformations to reform B-DNA. Through these structural transitions, flipons act as catalysts to frame transitions in chromatin structure. The scheme permits the reuse of the same set of cellular machines to generate a diverse set of transcriptomes just as the specific structure formed by the proper pairing of a codon triplet with its cognate tRNA enables a generic ribosome to produce a panoply of proteins [21]. Such programs run without changes to either protein or DNA sequence. By coordinately regulating flipon conformation, small RNAs can alter developmental outcomes. The increased phenotypic diversity present in offspring enhances the probability that a species propagates [62]. Interestingly, miR-744, miR-423 and miR-125a are highly expressed in the extraembryonic endoderm that interacts with the epiblast to shape early neural development [63]. By setting promoter state, these c-miR can potentially initiate feedback interactions between the endoderm and the epiblast in “a give-and-take” fashion to create gradients that initiate or inhibit the establishment of region-specific, cell-autonomous, developmental programs.

There are 40–1500 miR copies of each expressed miR per cell [64], a number sufficient to cover the available c-mRS sites within the 15–25,000 flipons of each type identified in Kouzine et al.’s data, assuming about half the miR are nuclear [8]. The outcome can vary depending on how many heptamers with an mRS are sufficiently open to allow for docking of an miR. Of the 16,386 possible heptamers, only 1185 have more than 20,000 copies within PP. Many of the most frequent heptamers are in simple repeats. For example, the d(TGTGTGT) heptamer is counted 183,958 times in PP, but maps to only 30,182 flipons as they each contain multiple copies of the heptad sequence. The remaining possible heptads are of lower frequency and allow for more sequence-specific targeting of flipons. Unlike cytoplasmic RNAs, where the stoichiometry is challenging [65], many potential binding sites for an RNA effector are buried within duplex DNA or cloistered within chromatin. MJ-matching c-miR can then exploit this variation to initiate strand-specific formation of alternative NoBs (Figure 1C,D). Flipon conformation will also depend on other types of *trans*-RNA [2]. In addition to AGO1 and AGO2, interactions may involve other argonaute family members, such as the PIWI proteins [9], that transfer miR from the cytoplasm to the nucleus in a regulated manner. The protein domains unique to each family member allows targeting of specific cellular machines to the flipons bound by their guide RNA [66,67]. We are presently investigating these possibilities.

The current findings lead to a general model where flipon conformations are modulated by miR (Figure 1). The combination of miR expressed in cells then shapes the genome, directing the readout of genetic information [68]. The enrichment of G and Z flipons in PP then provides a mechanism for fine tuning gene expression, with the miR interactions specifying where and when NoBs in a gene form. As illustrated for *Adgrb2* (Figure 10), G4 formation by a G flipon present in a promoter may be enhanced by one miR interaction and inhibited by another. The readout from this gene can then be tuned by the altering the miR manifest within a cell to change promoter shape. Similarly, the conformation of a G flipon in a gene body can also be regulated by an miR, as suggested by the interaction of miR-330 with the first intron of *Adgrb2* (Figure 10). By promoting G4 formation, miR-330 may force an RNA polymerase to fall off the DNA. The abortive transcript produced is likely rapidly destroyed. Such a mechanism acts to counter the permissive and pervasive transcription observed during different stages of metazoan development by the early termination of pre-mRNA elongation [69]. The close proximity of flipons to each other allows a single miR to regulate the conformation of both flipons, as illustrated by miR-744 interaction with *Acvr2b* (Figure 9C). In other cases, the outcome may depend on different miR to regulate the conformation of adjacent flipons. For *Grb10* (Figure 11), miR-210 and miR-187 target adjacent Z flipons. It is likely that these flipons compete for the available energy, with the formation of Z-DNA by one precluding formation by the other. The expression in a cell of miR specific for each flipon can determine the outcome with a flip by one, or the other, or by neither, possible. Similarly, miR allow the conformation of the neighboring G flipon to differ. One G flipon may form G4 because an miR binding to the C-rich strand is expressed while the flip of another is prevented by an miR that binds the flipon G-rich strand. While all G flipons in the experimental dataset can be targeted when all known miR are considered, only about 90% of Z flipons have mRS (Figure 6B). Perhaps Z flipons are also targeted by other classes of non-coding RNAs or, instead, they may be used only dynamically to buffer superhelical stresses within a topological domain.

While further experiments are required to elaborate our findings, support for the model comes from a number of existing studies that explore the effects of promoter targeted miR on gene expression. These include the reported upregulation by hsa-miR-324-3p of *RELA* expression in the pheochromocytoma PC-12 cell line that leads to an increased production of the pro-apoptotic caspase-3 protein. Mutations to either the promoter binding site or to the mRS decreased expression of luciferase reporter constructs, as did antagomirs specific for hsa-miR-324-3p. An association of AGO2 with RNA polymerase and TWIST1 induced by hsa-miR-324-3p was revealed using biotinylated miR-324-3p to pull down DNA-bound chromatin. Both gene expression and complex formation were diminished by siRNAs specific for AGO2 [70]. The hsa-miR-324-3p binding site identified in the study overlapped a site of predicted Z-DNA formation (Figure 12A,C). In this case, docking of the miR likely inhibits Z-DNA formation, making available the free energy needed to power the formation of transcriptionally competent complexes. Another study revealed that hsa-miR-744 stimulated expression of the *ARHGAP5* gene by binding to a promoter site -508 bases from the TSS, but not to a site -200 bases away. The -508 site overlaps a predicted Z-DNA forming sequence (Figure 12B,D) while the -200 site does not. When overexpressed in 5–8F nasopharyngeal cancer xenografts, hsa-miR-744 induces *ARHGAP5* expression to promote tumor growth and metastasis [71].

Other roles exist in the embryo for c-miR. Those targeting flipons in TRE protect against the spread of these ERE. The strategy exploits a vulnerability created by the AT-rich segments essential for copying and pasting new TRE back into the genome. We show that the SIDDs needed for these processes are targeted by a plethora of c-miR capable of suppressing their expression and splicing (Figure 2A). Rather than marked by cCRE, these sites are associated with H3K9me3 [72]. In such a context, SIDDs suppress G and Z flipons in nearby TRE promoters. The subset of miR targeting TRE SIDDs likely arose from the inverted repeats formed by insertion of two or more retrotransposons in reverse orientation adjacent to each other [73]. By preventing TRE reactivation, the miR likely serve as tumor suppressors. Such a role has been reported for miR-203a, with C57BL/6 miR203a null mice phenotypically normal but susceptible to Hras^G12V^-promoted papilloma formation [74]. Additionally, disruption of the miR-producing machinery by pathogens may relieve the suppression, leading to TRE overexpression and activation of innate interferon responses and Z-RNA-induced inflammatory cell death [75]. The TREs embedded in the genome then protect the host against invasion by new intracellular pathogens. Any novel TRE that gains a foothold in the genome can spread until targeting by miR becomes robust enough. In contrast, suppression by sequence-specific Krüppel-associated box domain zinc finger proteins takes millions of years to evolve [76]. During the period that ERE actively transpose, they likely spur species-specific changes by altering the expression of existing genes [77] and by redefining boundaries that separate topological domains [78,79] in a RNA-dependent manner [30,80]. The complexity of the relationship between host and repeat elements is further exemplified by LTRs that use transfer RNAs (tRNAs) to prime their reverse transcription and the deployment by the host of 3′ tRNA fragments to inhibit this step in ERE transposition [81].

The results we present are for a subset of flipons experimentally validated in a single cell line and likely underestimate the true extent of interactions with small RNAs that involve other AGO family members such as PIWI [9] and other types of *trans*-RNA [2]. Further, the detection of NoB by KMnO_4_ footprinting has limitations. This technique relies on thymine modification missing other unpaired bases such as guanine that can be detected with kethoxal [82]. Nevertheless, the evidence for regulation of flipon conformation by miR we present is compelling, far exceeding the 5-sigma discovery threshold applied in other physical sciences. Chemical footprinting techniques can be applied to any cell line and also to any stage of embryonic development to provide more information on how interactions of miR with flipons affect the readout of genetic information. The ability to control exposure and duration to the chemical [36] permits a careful mapping of chronological events to establish cause and effect. Such studies will address questions on how miR coordinate nuclear and cytoplasmic gene expression and provide further insights into the way nascent sense and anti-sense RNA transcripts, either coding or noncoding, regulate the interaction of miR with DNA flipons (Figure 1D). Therapeutic applications include the targeting of flipon sequences to either enhance or suppress gene expression [83]. The saRNA discovered previously by empirical means likely dock at open regions of the DNA helix produced either when both sense and anti-sense strands are transcribed or during DNA replication. In both cases, alternative DNA conformations may be transiently induced and catalyze the assembly of protein complexes at that location [31]. Indeed, the CpG island bound by the *CEBPA* saRNA contains many experimentally validated Z flipons [20,36]. Targeting of other small RNAs beyond the c-miR evaluated here and of other flipons classes warrants further investigation as do other known mechanisms for opening up the DNA double helix to enable switching of genetic programs by sequence-specific *trans*-RNAs. The tissue-limited expression of many small RNAs suggests that therapeutic mimetics or antagonists can be administered systemically, but act locally. Use of agents targeted at the highly variable flipon junction sequences will improve the specificity of their action. The identification of c-miR that are master regulators of other miR will enable the development of therapeutics with large effect sizes. It is likely that small RNAs regulate formation of all non-B-DNA structures.

## 7. Methods

### 7.1. Data Sources

KMnO4 mapping of non-B-DNA structures in murine lipopolysaccharide activated spleen B Cells (https://www.ncbi.nlm.nih.gov/CBBresearch/Przytycka/index.cgi#nonbdna (accessed on 21 December 2022)); c-mRS (Bartel [6]); human gene annotation GRCh37.87 from ENSEMBL (https://useast.ensembl.org/Mus_musculus/Info/Index, accessed on 5 May 2022), mouse basic gene annotation vM25 from GENCODE (https://www.gencodegenes.org/ accessed on 4 March 2022), mouse genome assembly GRCm38, candidate *cis*-Regulatory Elements and RepeatMasker tracks from UCSC ( https://genome.ucsc.edu/, accessed 24 January 2022), miR sequences from miRbase (https://www.mirbase.org/, accessed 19 November 2022), AGO1 ChIP-seq (GSE174905) and AGO2 ChIP-seq (GSE136467).

### 7.2. Methods

#### 7.2.1. Data Cleaning and Visualization

To uplift mouse data from an mm9 assembly to mm10, we used the liftOver executable from the UCSC binary utilities library [84]. The bigBedToBed executable was also used for cCRE data format conversion. To handle the input genomic coordinates, we used bedtools 2.30, while, to manage the nucleotide sequence data, we utilized the Python library biopython 1.79. To map regions from the input data to genomic features, we used ChIPseeker and ChIPpeakAnno R libraries. Data visualization was carried out both in tabular and chart formats using the plotly 5.8.0 Python library. To visualize specific locations in the genome, we used the UCSC Genome Browser. The code is available at https://github.com/theopavlove/article_2023_conserved_microRNAs (accessed on 12 February 2023).

#### 7.2.2. Gene Enrichment Analysis

Gene lists were mapped to Gene Ontology terms using the Database for Annotation, Visualization and Integrated Discovery (DAVID) Resource (https://david.ncifcrf.gov/) (accessed on 24 December 2022) [75]. The DAVID Knowledgebase integrates multiple sources of annotation to provide a number of measures of statistical enrichment for gene products that are grouped by common features, pathways or disease processes. While we present results based on false discovery rate (FDR) (i.e., the probability that a feature classed as a true positive (TP) is actually a false positive (FP), defined as FDR = FP/(FP + TP)), other measures of statistical significance are viewable in the Appendix A.

#### 7.2.3. Z-DNA Prediction

The propensity of sequences to form Z-DNA was analyzed using the transformer algorithm implemented in the Z-DNABERT program that is described in a preprint available at https://doi.org/10.1101/2023.01.12.523822 (accessed on 12 February 2023).

#### 7.2.4. Flipon Matches with c-mRS

The c-mRS binding was computed as three separate categories. The first one is a direct motif overlap (M) between mRS and a whole flipon region. The second is a junction overlap (J) of mRS with a single nucleotide in a flipon’s motif sequence, meaning that for each flipon, there are only two possible places for such overlaps: 5′-end minus 6 bp and 3′-end plus 6 bp. The last one is a motif-junction overlap (MJ), which considers mRS binding to, at most, 6 nucleotides of a motif from both 5′ and 3′ ends, with the rest of the sequence being outside of a flipon frame.

#### 7.2.5. Flipon Matches with Other Features in mm10 and HG38 Genomic annotations

Each flipon was associated with multiple genomic data items. The genomic feature mapping was computed via ChIPpeakAnno algorithm, which resulted in the following peak categorization: Promoter (≤1 kb, 1–2 kb, 2–3 kb), Exon/Intron, 5′/3′ UTR, Downstream and Distal Intergenic associations. In order to calculate overlaps between flipons and both cCREs and LINE/LTR repeats, each flipon was extended by 200 bp on either side and then matched to each region of interest. Each flipon was then assigned to one of the following categories: cCRE only, LINE/LTR only, cCRE and LINE/LTR, CTCF only, cCRE and CTCF, LINE/LTR and CTCF, cCRE and LINE/LTR and CTCF.

#### 7.2.6. Sites Bound by Each c-miR by DNA Strand

For each of the two flipon categories—cCRE only and LINE/LTR only—the number of mRS was counted with strand orientation taken into account. For each miR family, the number of bound sites was calculated with respect to the RNA transcription direction.

#### 7.2.7. Gene Promoter Analysis

We counted flipons present in PP (±1 kb from TSS) after splitting the regions into 100 bp windows. Flipons that overlapped two windows were counted as present in both. The counts were then summed across every gene promoter.

#### 7.2.8. Distances from AGO Peaks to TSS

For the analysis of distance to TSS, we used the ENCODE annotation. Only the distance to the nearest region of interest (AGO1 or AGO2 ChIP peak) is shown on the plot (Figure 6G). A rolling average with a window of 8 bases is applied to smooth the curve.

#### 7.2.9. ChIP-seq Analysis:

For the analysis of AGO1 and AGO2, we used custom R 4.1.3 and Python 3.7.13 scripts. Overlaps were calculated using GenomicRanges R library, and values for smooth curves were calculated using the zoo R library. The distances to TSS were drawn using the ggplot2 R library. The scripts are available in the repository https://github.com/Madund3ad/article-mi-RNA/ (accessed on 17 February 2023).

#### 7.2.10. AGO Overlap Analysis

For the analysis of overlaps between ENCODE AGO narrow peaks, experimental Kouzine SIDD peaks and promoters, we took 1 kb upstream regions from TSS as a promoter. TSS coordinates were obtained from the ENCODE annotation. Two regions were considered to be overlapping if they had at least one base pair in common. Significance of the overlap was calculated using Monte-Carlo simulation (n = 1000).

## Figures and Tables

**Figure 1 ijms-24-04884-f001:**
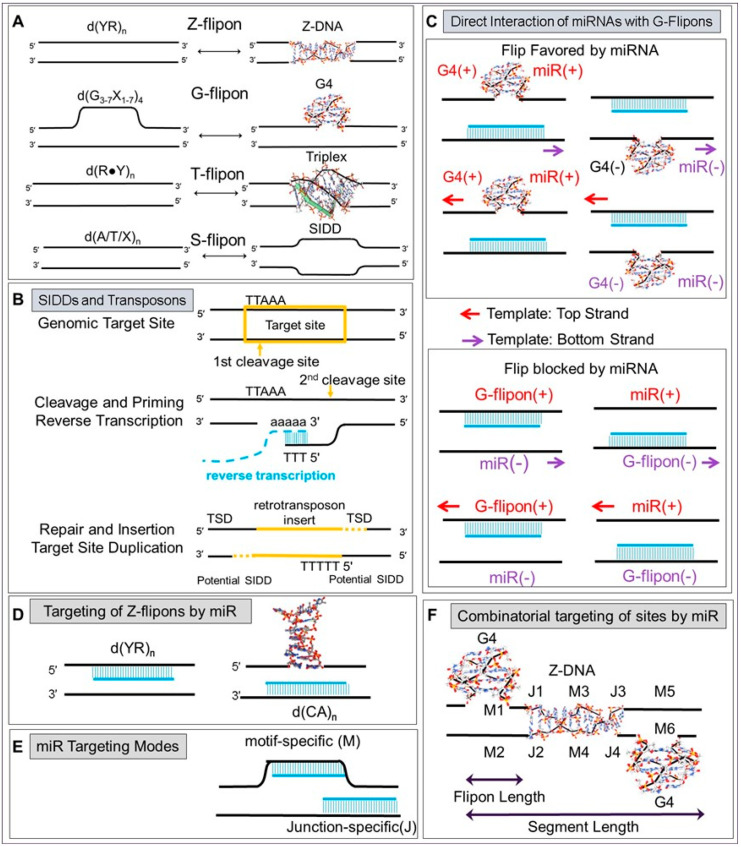
DNA and RNA flipons. (**A**) Examples of repeat sequences forming alternative DNA conformations. Z flipons with alternating pyrimidine (Y) and purine motifs (R) invert DNA bases to form left-handed Z-DNA; G flipons fold one strand into a quadruplex (G4) formed from G tetrads; triplexes widen the major groove to bind an additional nucleic acid Y or R strand (shown as a green tube) and can form from a local fold of one DNA strand onto the other (H-DNA) or by incorporation of a third nucleic acid strand produced in *trans*; S flipons undergo stress-induced DNA destabilization (SIDD) to form single-stranded regions. (**B**) The prevalence of S flipons in the genome is increased by retrotransposition of LINE elements that depend on AT-rich sequences for reinsertion of their RNA into the genome. The process results in target-site duplication (TSD). (**C**) An example showing how miR sequence matches with G flipons can promote or prevent formation of G4. If the G-rich strand and the miR have a matching sequence (either “+,+” or “−,−”, i.e., labels for both G4 and miRNA are the same (either both colored red or both colored purple), then G4 formation is promoted. However, if the miR sequence match is with the opposite strand (“−,+”, or “+,−”, i.e., the labels for G-rich strand and the miR have a different color), then G4 formation is suppressed. The outcome does not depend on the direction of RNA transcription (the red arrow indicates that the top strand is the template strand, while the arrow is colored purple when the bottom strand is used for the template). G4 can form on the coding strand when the arrow, G4 and miR labels have the same color. Otherwise, G4 forms on the template strand when the G4 and miR labels have the same color but the arrow color is different. While flipons on the coding strand can initiate assembly of complexes at promoters, G4 formed within the 5’ UTR on the template strand close to the TSS can cause polymerases to fall off, resulting in premature transcription termination in a mRS dependent manner. (**D**) miR will usually inhibit Z-DNA formation. However, in sequences with a d(GT)_n_-rich strand, the formation of non-canonical wobble d(G:T) base pairs may lead to formation of a Z-DNA hairpin [24]. With longer repeats (>11 bases), a Z quadruplex may form instead [26]. (**E**) The greater sequence variation at junctions (J) allows for specific targeting of particular flipons by miR. The miR-recognizing flipon motifs (M) have more global effects on transcription as they target more RNAs. (**F**) Flipons of different types can colocalize. The conformation adopted by each flipon depends upon the set of miR present in a cell. Competition with other nearby flipons for the locally available free energy that arises from the generation of negative supercoils within the region further influences promoter shape. Both the sequence composition and the flipon length affect the outcome of these interactions. The particular NoB formed at a specific location in the genome also can vary by context, reflecting the cellular manifest of miR expressed at that moment in time and space (M: motif binding miR; J: junction binding miR; TSS: transcription start site).

**Figure 2 ijms-24-04884-f002:**
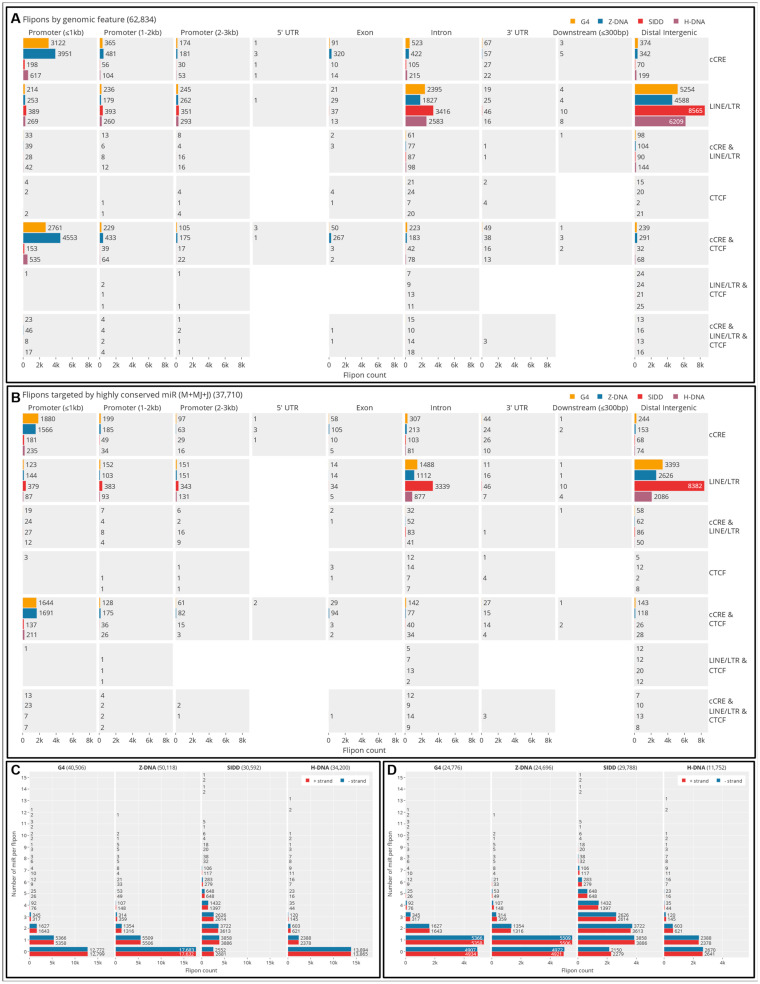
Flipon matches with c-mRS. (**A**) Flipon counts by flipon type and by genomic feature grouped by the association with either ENCODE cCRE or TRE. (**B**) Counts for those flipons with perfect sequence matches to a c-mRS. (**C**) Number of c-mRS matches per flipon for each flipon type, where zero means that there is no match. (**D**) Count of flipon matches with mRS for each DNA strand, where zero means a strand has no c-miR binding site. Results are relative to the direction of transcription (red denotes the coding strand, labeled as “+”, while blue is the template strand, labeled as “−”). Each match for a c-miR is counted separately: a c-miR may have multiple matches on one strand and zero on the other (cCRE: candidate *cis* regulatory regions; LTR: long terminal repeats; LINE: long interspersed nuclear elements; TRE: transposon repeat elements; c-mRS: conserved miRNA seed sequence).

**Figure 3 ijms-24-04884-f003:**
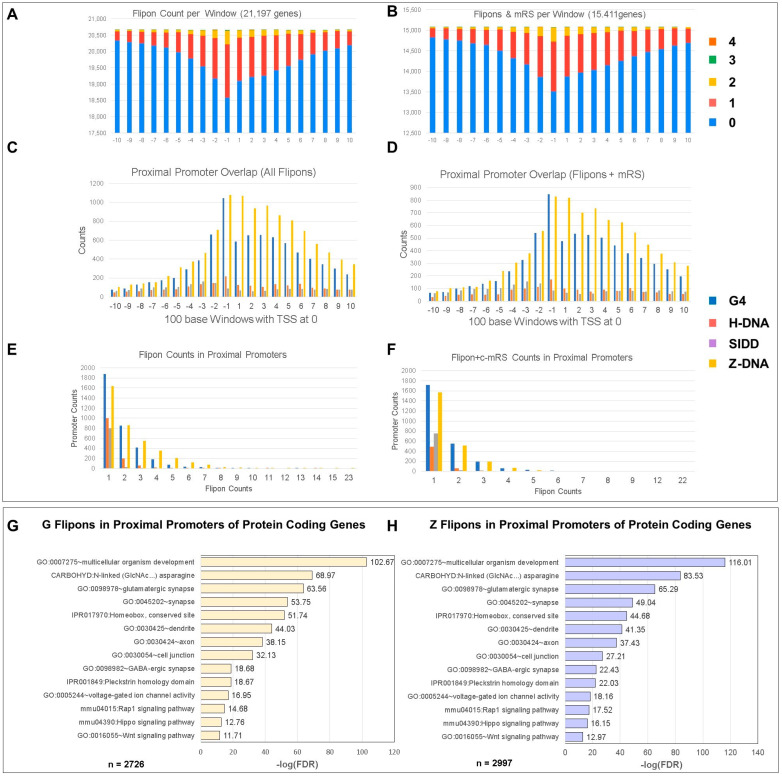
Flipons and c-mRS in PP. (**A**) The number of flipons present in each 100 bp window on either side of the TSS (centered at “0”). (**B**) The number of c-mRS in 100 bp windows either side of the TSS. (**C**) The count of different flipon types in PP in 100 bp windows centered on the TSS. (**D**) The number of mRS matches with flipons in PP in 100 bp windows centered on the TSS. (**E**) PP contain a variable number of flipons. (**F**) PP also vary in the number of flipons they have that bind c-miR. (**G**,**H**) Gene Ontology processes enriched for genes with G or Z flipons in their PP that match c-mRS. The letter “n” at the bottom of (**G**,**H**) refers to the number of DAVID IDs submitted for each analysis (mRS: miR seed sequence; TSS: transcription start site).

**Figure 4 ijms-24-04884-f004:**
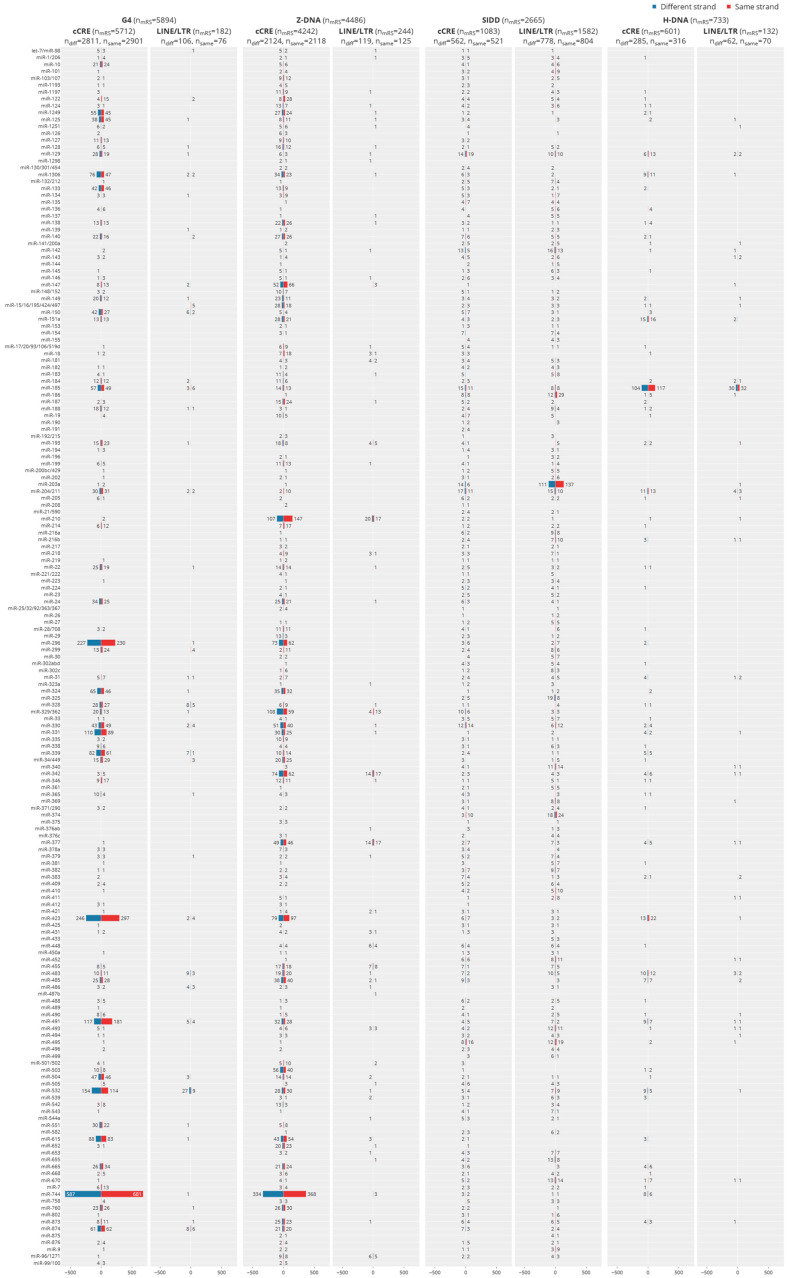
Number of experimentally validated flipon sites bound by each c-miR for each DNA strand relative to the coding strand. Red indicates that c-mRS are on the coding strand while blue indicates that c-mRS are on the template strand. Sites with cCRE marks and LTR/LINE repeats are shown for each flipon type (cCRE: candidate *cis*-regulatory elements; LTR: long terminal repeats; LINE: long interspersed nuclear elements).

**Figure 7 ijms-24-04884-f007:**
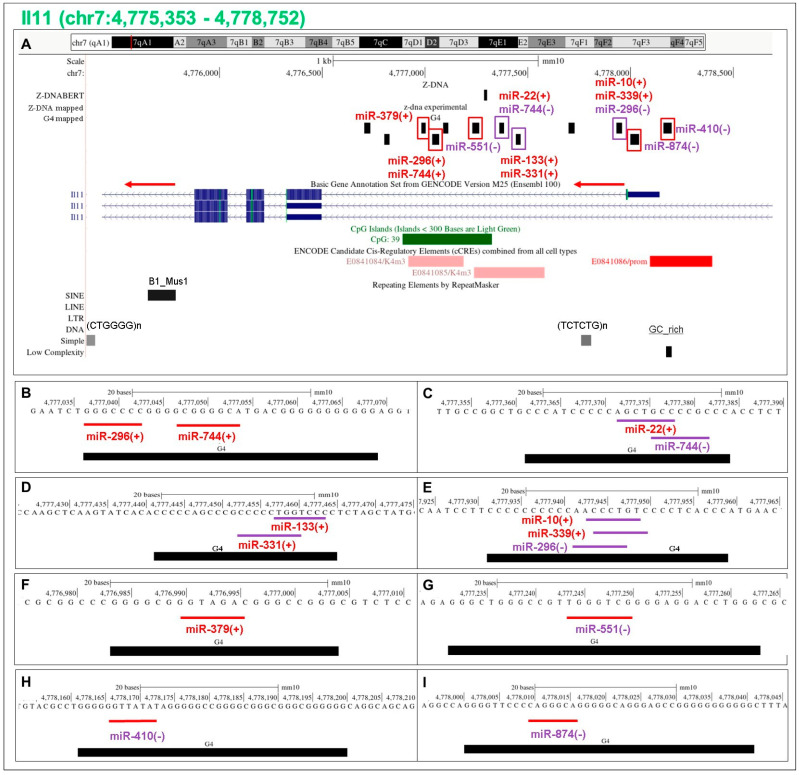
The promoter of the mouse interleukin 11 (*Il11)* (**A**) Location of experimentally validated and predicted flipons (shown as black rectangles) [36] (**B**–**I**) Mapping of c-mRS overlaps. The following line coloring scheme is used to indicate the flipon strand that is prone to form a NoB conformation: G flipons: red for top strand, purple for lower strand; Z flipons: blue for top and green for lower strands; S flipons: brown for top and orange for bottom. The direction of the arrow drawn over the transcript indicates that the top strand is the template strand. We color the arrow red in this case, but in other cases, the arrow is colored purple to indicate that the bottom strand is the template. The miR name is also color coded to indicate the strand that matches the seed sequence: red if the mRS match is with the top DNA strand; purple if the mRS match is with the bottom strand. In addition, “+” indicates a match with the top strand and “−” with the bottom strand. In the case of G flipons, a color match (equivalent to the “+,+” shown in Figure 1) indicates that the miR sequence promotes G4 formation by the “+” strand, while a “−,−” color match indicates G4 formation occurs on the ‘−‘ strand. A color mismatch (i.e. ”−,+” or “+,−”) indicates that the miR prevents G4 formation by the flipon. The cCRE elements and repeats present in the regions are also displayed. The lower panels show the precise sequence alignment between c-mRS and flipons (mRS: miRNA Seed Sequence).

**Figure 8 ijms-24-04884-f008:**
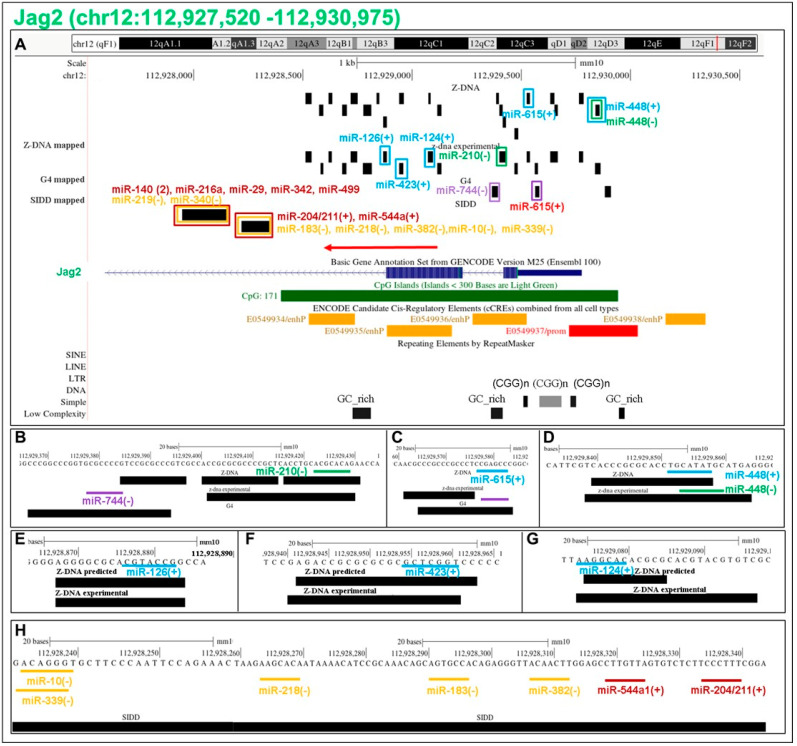
Location of experimentally validated flipon and miR sequence overlaps in the promoter of *Jag2*. (**A**) The experimental Z-DNA data aligns with Z-DNA forming sequences predicted by the Z-DNABERT program. The miR and the flipons (located within a box in the upper panel and underlined in the lower panel) are color coded to show whether they are present on the top or bottom DNA strand (see Figure 7 legend). The color of the miR name is similarly color coded. In the case of G flipons, a color match (either “+,+” or “−,−” as shown in Figure 1) indicates that the miR sequence promotes G4 formation, while a color mismatch indicates that the miR prevents G4 formation. In all the cases shown, the miR prevents Z-DNA formation. The red color of the arrow indicates that the top strand is the transcription template. The cCRE elements and repeats present in the regions are also displayed. The lower panels (**B**–**H**) show the precise sequence alignment between miR and flipons (mRS: miRNA Seed sequence).

**Figure 9 ijms-24-04884-f009:**
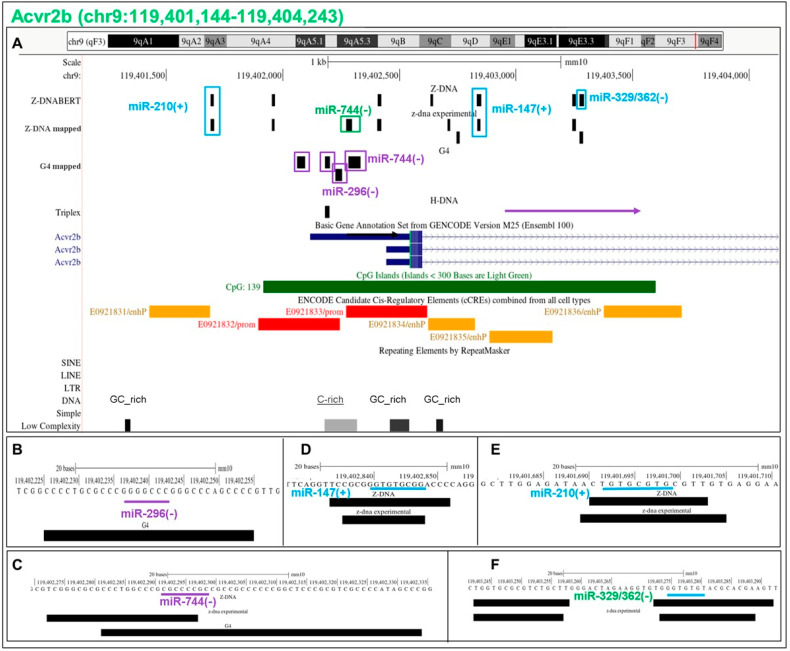
Flipons in the *Acvr2b* promoter. (**A**) The flipons are indicated by black boxes. They are color coded by the strand that contains the flipon motif, as shown by the boxes surrounding each flipon (panel **A**) or a line under its sequence (panels **B**–**F**). The strand matching the mRS is indicated by the color of the miR lettering (as described in the legend to Figure 7). The color of the arrow indicates whether the top or bottom strand is the transcription template. For G flipons, G4 formation occurs on the template strand when all the colors of the flipon boxes, miR and arrows match. Each flipon class is colored differently to help distinguish them from each other: G: red/purple; Z: green/blue; S flipons: brown/orange. cCRE marks are colored red to indicate a promoter or orange to indicate enhancers. (**B**–**F**) Mapping of mRS to flipon sequences. A match with the top strand is also indicated by “+” and to the lower strand with “−” (mRS: miR seed match).

**Figure 10 ijms-24-04884-f010:**
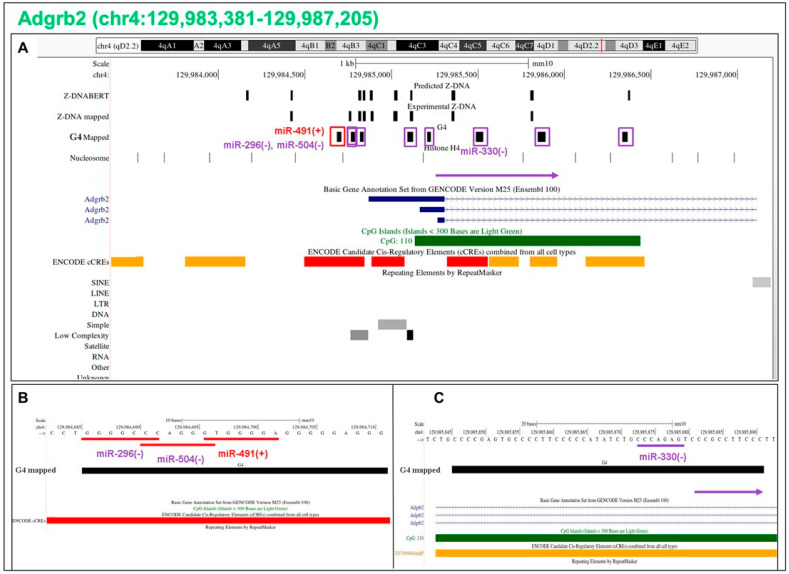
Flipons in the *Adgrb2* promoter. (**A**) Coloring is as described for Figure 7. The promoter region is also characterized by the nucleosome-free region (NFR) that is mapped at nucleotide resolution through hydroxyl radical footprinting with a cysteine-modified histone H4 [50] and shown in the nucleosome track. (**B**,**C**) Mapping of miR to flipon sequence is shown. A match with the top strand is also indicated by a “+” and with a match to the lower strand marked with a “−”.

**Figure 11 ijms-24-04884-f011:**
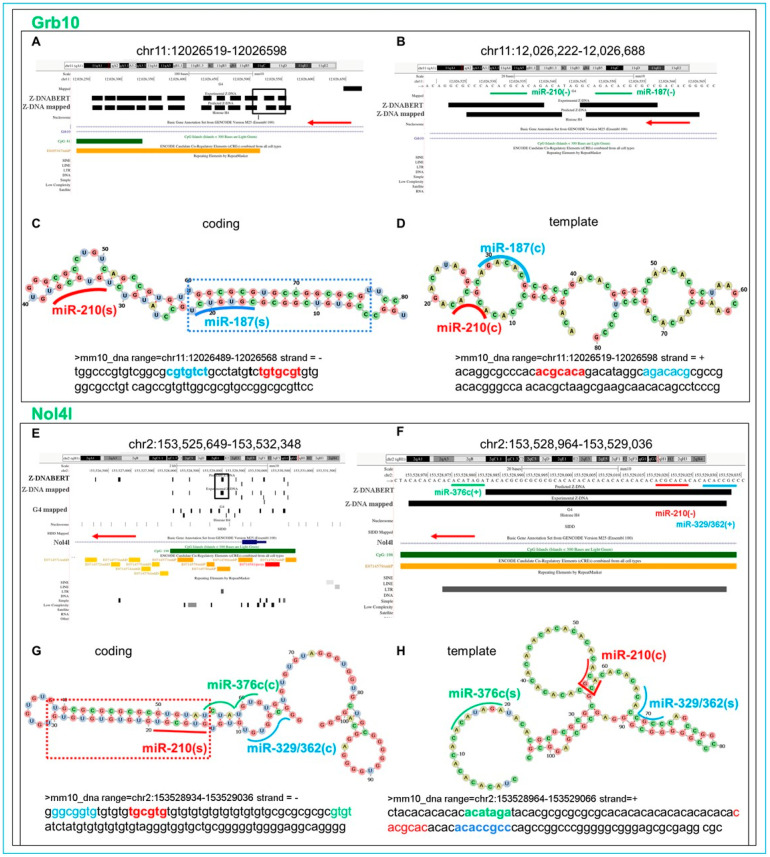
Z flipons and different roles for miR. (**A**) Location of Z flipons in the *Grb10* promoter. (**B**) Mapping of miR to the flipon boxes. (**C**,**D**) Folding of template and coding strands with perfect matches to c-mRS (denoted by “(s)” for seed after the miR name) or to their base complement (indicated by “(c)“ appended to the miR name). The colored line overlying the miR (c) and (s) sites mirror the text coloring in the boxes underneath each figure. (**E**) Location of Z flipons in the *Nol4l* promoter. (**F**) Mapping of mRS to the Z flipon boxes (**E**,**G**,**H**) Folding of *Nol4l* template and coding strands with both c-mRS matching (s) and complementary (c) sequences mapped. In panels (**C**,**G**), the dotted boxes encase a potential Z-DNA hairpin. The GT-rich *Nol4l* coding sequence could also fold to form a DNA quadruplex using noncanonical basepairs [24,26] (c-mRS: conserved miR seed match).

**Figure 12 ijms-24-04884-f012:**
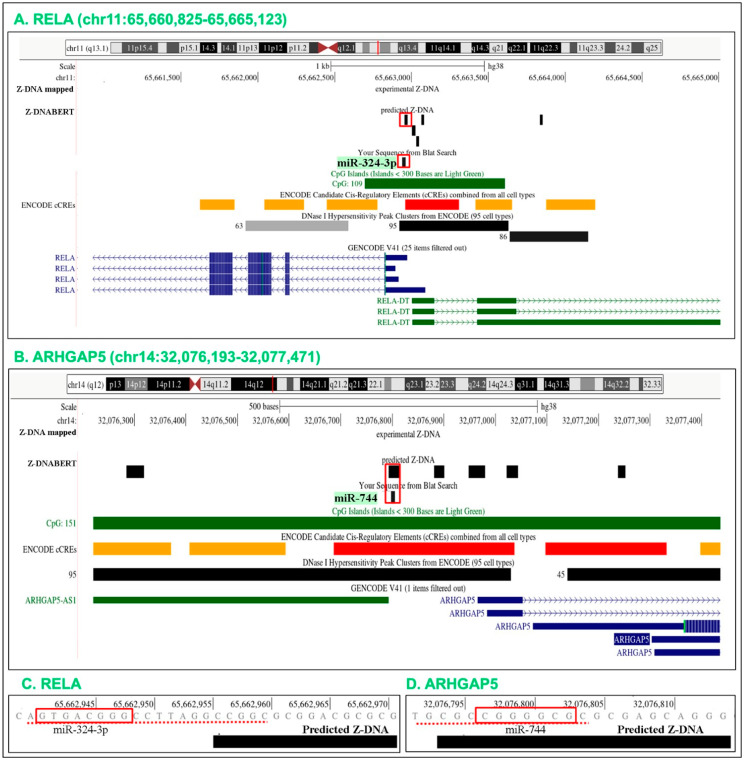
Mapping to Z flipons of c-miR experimentally shown to regulate human gene expression. (**A**) hsa-miR-324-3p targets the promoter of *RELA* [70]. (**B**) hsa-miR-744 docks to the *ARHGAP5* promoter [71]. The overlapping miR and Z flipons are boxed in red. The *RELA* (**C**) and *ARHGAP5* (**D**) seed sequences are within the boxes. The *RELA* sequence is that for the bottom strand (i.e., in the 3’ to 5’ direction). The red dotted line indicates additional overlap of miR-324-3p and miR-744 with the Z flipon in each gene.

## Data Availability

The data is available at websites provided in the method section or in the Appendix A.

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
