# Peer review of "Conserved microRNAs and Flipons Shape Gene Expression during Development by Altering Promoter Conformations"

_ijms, 2023, doi:10.3390/ijms24054884_

Round 1
Reviewer 1 Report
Most models, ideas, and hypotheses around non-canonical DNA-structures arose in the 1980s and 90s of the last centuries. As it often happens in experimental sciences, the field was given up by almost all of us, because methods, technologies, and approaches at that time weren't suited to establish causal evidence for biological functions.
Only a couple of years ago, when a paper on the functions of ADAR1 and ZBP1 in genetically triggered immunological diseases was published - and the authors weren't even aware of Z-DNA or Z-RNA binding motifs in these proteins - it was Alan Herbert who lead the field to an impressive, top level renaissance. He also invented the term "flipon" as a dynamical equivalent of the static "codon", not only for Z-NA, but also for other equally important, torsional strain-induced nucleic acid structures.
In recent months, an impressive number of high impact papers generated the causal evidence on the biological functions of Z-NA, which we were not able to establish in those "old days". Since I was involved as a reviewer frequently, I was a bit reluctant to accept also this manuscript for review. This may however be my personal "conflict of interest" by an indirect mechanism...
The current scientific view on the functional dynamics of „flipons" in chromatin is nicely summarized by citing from one of the most recent papers of Herbert et al: "In such scenarios, nucleosomes provide a way to store the energy required for remodeling of the chromatin landscape. They serve as batteries to power DNA transactions. In these circuits, flipons function as capacitor or resistor elements to smooth the energy transfer from source to sink."
I wouldn’t give a summariz.ing review of the authors’ findings here. But in essence, the bioinformatically highly significant combination of miRNA sequences and flipons in the proximal promoters of a set of cell-fate determining genes is shown, as well as in a second set miRNA and flipon combinations, essential for retrotransposon maintenance. Experiments in molecular biology and bioinformatic analyses were both employed to reveal the significance of scientific claims. Though bioinformatics provides correlative evidence only, the levels of statistical significance reached, may serve as a substitute for causal evidence.
This is a close to perfect paper - with respect to presentation of bioinformatic results as well as in terms of concise argumentation!
p.s. the only sentence, I don't like, is the first one of the "Abstract". Even in prokaryotes the interaction between gene expression regulating proteins and DNA is definitely not "sequence specific". The 4-letter nucleic acid "language" and the 20 (21)-letter protein language always speak through structure::structure interactions, like (in DNA) inverted repeats, cruciforms, palindromes, etc.
Author Response
Thank-you very much for your review and your suggestion. We changed the first line of the abstract and added a reference to the experimental data we found to support that the interactions between miRNAs and flipons are indeed causal. Here is the new abstract
"The classical view of gene regulation draws from prokaryotic models where responses to environmental changes involve operons regulated by sequence-specific protein interactions with DNA, although it is now known that operons are also modulated by small RNAs. In eukaryotes pathways based on microRNAs (miR) regulate the readout of genomic information from transcripts, while alternative nucleic acid structures encoded by flipons influence the readout of genetic programs from DNA. Here we provide evidence that miR and flipon-based mechanisms are deeply connected. We analyze the connection between flipon conformation and the 211 highly conserved human miR that are shared with other placental and other bilateral species. The direct interaction between conserved miR (c-miR) and flipons is supported by sequence alignments and the engagement of argonaute proteins by experimentally validated flipons as well as their enrichment in promoters of coding transcripts important in multicellular development, cell surface glycosylation and glutamatergic synapse specification with enrichments significant at false discovery rates as low as 10-116. We also identify a second subset of c-miR that targets flipons essential for retrotransposon replication, exploiting that vulnerability to limit their spread. We propose that miR can act in a combinatorial manner to regulate the readout of genetic information by specifying when and where flipons form non-B DNA (NoB) conformations, providing the interactions of the conserved miR-324-3p with RELA and the conserved miR-744 with ARHGAP5 as examples."
Reviewer 2 Report
The study on miRNAs and Flipons are interesting and explains the connection between flipons and human miRs. the authors have nicely pictured the results of miRs and flipons.
Some of the minor concerns include the following:
In methods section, there are sub-headings viz., Data, methods and analysis which is a bit unclear. I suggest to modify methods section and merge with sub-headings.
Please see attachment for additional comments:

Author Response
Thanks for your review and suggestions. We elaborated further on the questions posed.
- Other instances where small RNAs may modulate gene expression by interaction with DNA. We added the following text in the introduction:
"Small synthetic RNAs (saRNA) can also activate gene expression in an AGO2 dependent fashion. The first saRNA were initially discovered empirically by tiling of the progesterone receptor (encoded by PGR) promoter, excluding regions with either high GC content or low sequence complexity [17, 18]. Later, the p21 promoter (encoded by CDKN1A) [19] and the CCATT/enhancer binding protein alpha (encoded by CEPBA) were also effectively activated by saRNA [20]. In each of these three cases, the target site in the promoter of each gene is within a region of antisense transcription (PGR-AS1 for PGR, DINOL for CDKN1A and the polycomb-associated non coding RNA ENST00000587312.1 for CEPBA).However, targeted destruction of PGR-AS1 with siRNAs does not diminish saRNA induced gene expression [17], while siRNAs against the anti-sense transcript increase anti-sense CEBPA RNA transcription [20]. The PGR saRNA binds within a LIMB8 LINE (long-interspersed nuclear element) while the CEPB saRNA site lies within a CpG island, raising the question whether such classes of repeat sequences are also targeted by cellular small RNAs to regulate gene expression."
and this text to the discussion
"Therapeutic applications include the targeting of flipon sequences to either enhance or suppress gene expression [76]. The saRNA discovered previously by empirical means likely dock at open regions of the DNA helix produced either when both sense and anti-sense strands are transcribed or during DNA replication. In both cases, alternative DNA conformations may be transiently induced and catalyze the assembly of protein complexes at that location [27]. Indeed CpG island bound by the CEBPA saRNA contain many experimentally validated Z-flipons [20, 33]. " - We have not finished our analysis of other classes of small RNAs and have added the following note in the discussion
"Flipon conformation will also depend on other types of trans-RNA [2]. Besides AGO1 and AGO2, interactions may involve other AGO family members, such as the PIWI proteins [9] that transfer miR from the cytoplasm to the nucleus in a regulated manner [60, 61]. We are presently investigating these possibilities." - We reformatted the headings in the methods section and described the DAVID resource more fully
"While we present results based on false discovery rate (FDR) (i.e. the probability that a feature classed as a true positive (TP) is actually a false positive (FP), defined as FDR = FP / (FP + TP)), other measures of statistical significance are viewable in the Supplementary Data."